# Form factor determination of biological molecules with X-ray free electron laser small-angle scattering (XFEL-SAS)

Clement E. Blanchet [1,14 ✉], Adam Round [2,14 ✉], Haydyn D. T. Mertens [1], Kartik Ayyer [3,4], Melissa Graewert [1], Salah Awel[5], Daniel Franke[1,13], Katerina Dörner[2], Saša Bajt [3,5], Richard Bean[2], Tânia F. Custódio[1,6], Raphael de Wijn [2], E. Juncheng [2], Alessandra Henkel[5], Andrey Gruzinov[1], Cy M. Jeffries [1], Yoonhee Kim[2], Henry Kirkwood [2], Marco Kloos [2], Juraj Knoška[5], Jayanath Koliyadu [2], Romain Letrun [2], Christian Löw [1,6], Jana Makroczyova[2], Abhishek Mall[4], Rob Meijers[7], Gisel Esperanza Pena Murillo [5], Dominik Oberthür [5], Ekaterina Round[2], Carolin Seuring [6,8,9], Marcin Sikorski[2], Patrik Vagovic[2,5], Joana Valerio[2], Tamme Wollweber[3,4], Yulong Zhuang[3,4], Joachim Schulz [2], Heinrich Haas [10], Henry N. Chapman [3,5,11], Adrian P. Mancuso[2,12] & Dmitri Svergun [1,13 ✉]

Free-electron lasers (FEL) are revolutionizing X-ray-based structural biology methods. While protein crystallography is already routinely performed at FELs, Small Angle X-ray Scattering (SAXS) studies of biological macromolecules are not as prevalent. SAXS allows the study of the shape and overall structure of proteins and nucleic acids in solution, in a quasi-native environment. In solution, chemical and biophysical parameters that have an influence on the structure and dynamics of molecules can be varied and their effect on conformational changes can be monitored in time-resolved XFEL and SAXS experiments. We report here the collection of scattering form factors of proteins in solution using FEL X-rays. The form factors correspond to the scattering signal of the protein ensemble alone; the scattering contributions from the solvent and the instrument are separately measured and accurately subtracted. The experiment was done using a liquid jet for sample delivery. These results pave the way for time-resolved studies and measurements from dilute samples, capitalizing on the intense and short FEL X-ray pulses.

[1] European Molecular Biology Laboratory EMBL, Hamburg Site, c/o DESY Notkestrasse 85, 22603 Hamburg, Germany. [2] European XFEL GmbH, Holzkoppel 4, 22869 Schenefeld, Germany. [3] The Hamburg Centre for Ultrafast Imaging, Universität Hamburg, Luruper Chaussee 149, 22761 Hamburg, Germany. [4] Max Planck Institute for the Structure and Dynamics of Matter, Luruper Chaussee 149, 22761 Hamburg, Germany. [5] Center for Free-Electron Laser Science CFEL, Deutsches Elektronen-Synchrotron DESY, Notkestr. 85, 22607 Hamburg, Germany. [6] Centre for Structural Systems Biology (CSSB), Notkestrasse 85, D-22607 Hamburg, Germany. [7] Institute for Protein Innovation (IPI), 4 Blackfan Circle, Boston, MA 02115, USA. [8] Department of Chemistry, University of Hamburg, Hamburg, Germany. [9] Leibniz Institute of Virology, Hamburg, Germany. [10] BioNTech SE, 55131 Mainz, Germany. [11] Department of Physics, Universität Hamburg, Luruper Chaussee 149, 22761 Hamburg, Germany. [12] Department of Chemistry and Physics, La Trobe Institute for Molecular Science, La Trobe University, Melbourne, Victoria 3086, Australia. [13] Present address: BIOSAXS GmbH, Notkestr. 85, 22607 Hamburg, Germany. [14] These authors contributed equally: Clement E. Blanchet, Adam Round. ✉email: clement.blanchet@embl-hamburg.de; adam.round@xfel.eu; dmitri.svergun@embl-hamburg.de

X-rays delivered by free-electron lasers offer great experimental opportunities for structural biology. Already employed for serial crystallography and single-particle imaging, the short and intense XFEL pulses are particularly relevant for X-ray scattering studies of bio-molecular reactions and conformational transitions, providing data with an unmatched time resolution[1,2].

In small-angle X-ray scattering (SAXS), X-rays scattered by the electrons within a sample are analyzed to gain insight into states and processes at the nanometer scale[3]. Biological solution SAXS (BioSAXS) is routinely employed at synchrotrons to study biological molecules in solution to determine the scattering form factors and intermolecular structure factors of proteins, nucleic acids, and other biomacromolecules. Such insight can also be important for the characterization of nano-scaled bio-pharmaceutical products, consisting of proteins, nucleic acids, lipids, polymers, or other moieties, where internal organization and quality often depend in a complex manner from manufacturing and environmental parameters. Lipid nanoparticle products for delivery of messenger RNA for vaccination against infectious diseases are a prominent example that has gained great public attention recently[4,5].

The scattering form factor of a molecule provides valuable insights into the spatial distribution of its electrons, allowing one to deduce information about its size, shape, and conformational flexibility[6]. SAXS characterizes molecules in solution, providing structural information of lower resolution compared to techniques like cryo-electron microscopy (CryoEM) or macromolecular crystallography (MX), which involve trapping molecules in ice or crystals. However, this lower resolution is balanced by the advantage of easily modifying the solution composition, enabling the probing of concomitant structural and dynamic changes in a wide range of biological systems. High-resolution structures and models obtained experimentally or computationally[7] are readily used to interpret SAXS data in hybrid modeling approaches[8,9]. SAXS is also routinely applied to the analysis of heterogeneous mixture and oligomeric/aggregation assembly processes that are challenging to study using other structural techniques. It is a particularly powerful tool for the study of dynamic/flexible systems such as intrinsically disordered proteins and disordered protein regions[10]. SAXS has garnered large interest within the scientific community, enabling researchers to gain unique insights into the structural properties of biomolecules in solution.

Standard solution scattering measurements of biological macromolecules are conducted regularly at optimized synchrotron beamlines around the world[11–18]. However, the relatively weak SAXS signal from biomolecules limits synchrotron BioSAXS to samples with fairly high concentrations above 0.1–1 mg/ml, a challenge for samples that are difficult to purify, such as membrane proteins and protein complexes. Thus, synchrotron BioSAXS analysis is not readily applicable when sample production and purification only provide a very dilute sample with the necessary stability. Of the photons counted on the detector, only a small fraction, i.e., about 1 out of a million[19], are scattered by the molecule of interest, with the majority scattered by the solvent/buffer or by beamline elements, which produce background. To obtain the target molecular form factor, the scattering curve of the background signal collected on the solvent/buffer alone is subtracted from the sample scattering curve, taking care to properly determine the instrument background and scattering from the measurement cell. This is routinely performed on laboratory sources and in synchrotron SAXS, where the accuracy of buffer subtraction is facilitated by automation[20] with measurement as close in time as possible with intensity scaling calibrated using the measured direct beam intensity.

The development of X-ray free-electron lasers (XFELs) offering intense femtosecond-duration pulses has paved the way for measuring macromolecular structures with serial femtosecond crystallography (SFX)[21,22]. In addition, this technology provides a solid platform for noncrystalline solution measurements of ultra-dilute samples (<0.1 mg/ml) that are not generally possible using current synchrotron sources. While solution measurements on biological molecules have been conducted using FELs, the form factors of biological particles have thus far only been determined for large supramolecular constructs/assemblies[23,24]. XFEL scattering experiments have been conducted on smaller protein systems[1,2,25], with the variations in the signal between samples in different conditions revealing important information on conformational switching caused by excitation events. The interpretation and modeling strategies for such experiments are limited by a multitude of assumptions, including confidence in a known starting structure and the reliability of additional a priori knowledge used. A robust approach for the direct extraction of the scattering form factor would thus be advantageous for biological measurements at XFELs, leveraging the advancements made on third-generation sources for the successful analysis of scattering data.

Due to the complexities inherent in XFEL data collection and the irreproducible nature of the probing X-rays at FEL sources (self-amplified spontaneous emission, SASE[26]), accurate background subtraction is non-trivial, and to our knowledge, the full form factor of a biological macromolecule in solution has, until now, not been reported. To extract the molecular form factor, any difference in the experiment between the sample and buffer measurement (jet and X-ray pulse stability) must be minimized. The use of standardization and automation for sample delivery, in combination with the high repetition rate of the European XFEL (MHz) enabling statistical averaging and outlier filtering, is particularly advantageous in this respect.

Direct duplication of synchrotron experiments for XFEL data collection is not feasible. A common sample environment for routine solution SAXS delivers a flowing liquid sample for exposure in a thin-walled quartz capillary (1–2 mm I.D.), held under vacuum. Such a setup is problematic at an XFEL. The power per XFEL pulse rapidly vaporizes the buffer solution, and the resulting expansion of newly formed gas would destroy the capillary. The XFEL beam itself is also likely to damage the capillary directly. A more appropriate strategy for liquid-based sample delivery at an XFEL is developed for serial crystallography and adapted for noncrystalline samples, i.e., a direct injection (jetting) of a liquid sample into an evacuated chamber for interaction with the beam. The most prevalent liquid jet method for low viscosity media accelerates the liquid using a gas sheath in the form of a gas dynamic virtual nozzle (GDVN)[27,28], enabling a sample replenishment (reestablishing the interrupted jet) between adjacent pulses. This acceleration enables liquid jets to operate above $40\,ms^{-1}$ and with sample replenishment at MHz repetition. This rate is needed to match the delivered pulses at the European XFEL[21,22]. We anticipated variations in the observed scattering due to a slightly different intersection of the X-ray pulse with the jet, as both the incident pulses[29] and the liquid jet[30] naturally vary in their physical properties as a function of time. The position of the X-ray on the jet will influence the incident angle and resulting scattering flares observed on the detector. The effective sample thickness varies with that of the jet and the jet alignment. We also anticipated possible differences in the jet parameters when the buffer is delivered with and without the sample. Such differences would lead to over (or under) estimation of the buffer scattering and to inaccurate form factor recovery. One further complication originates in the explosion of the jet by the first pulses of the train. Debris from the explosion,

still present around the sample, scatter X-rays from the subsequent pulses of the train. The spontaneous nature of XFEL pulse generation (SASE)[26] also contributes to the variability of the X-ray pulses and intensity. For these reasons, SAXS/WAXS experiments in solution at XFELs have thus far been limited to differential measurements. For example, difference curves can be obtained by collecting light and dark states alternatively, allowing one to obtain insight into the time-resolved behavior of the sample while minimizing the impact of these mentioned variations[1,2]. For direct form-factor recovery from XFEL-SAS experiments, complications arising from the variability of X-ray pulses and the stability of the liquid jet sample delivery must be overcome, and this forms the main goal of this work.

To overcome these limitations and recover protein form factors, several options were explored. The beam parameters (intensity, repetition rate) were adjusted together with the jet speed. The benefits and possible issues of an HPLC autosampler coupled with GDVN were evaluated. The autosampler allows one to inject samples directly into the buffer flow in a convenient and automated manner. It allows one to measure the buffer before and after the sample in a single run (without switching the reservoir). The autosampler is able to handle small volumes, and sample manipulation is greatly facilitated with volume reduced down to a few microliters. Furthermore, data collected at the XFEL are compared to control measurements of the same samples collected at a synchrotron beamline.

Well-characterized protein samples were used for the first tests to establish a measurement protocol. Bovine serum albumin (BSA), a 66 kDa protein derived from cows, is arguably the most measured protein across all BioSAXS beamlines. Predominantly monomeric in solution, it serves as a SAXS calibration standard for molecular weight estimation[31]. Apoferritin is a large protein construct (450 kDa) that facilitates controlled storage and delivery of iron *in corpore*. It consists of 24 subunits that assemble to form a hollow protein cage. The protein has diverse applications: it is used for drug encapsulation and delivery[32], as a precursor for the growth of carbon nanotubes[33], and as a nano-reactor for the synthesis of nanoparticles[34]. Ferritin functionalized with spike proteins is also being considered for the development of a single-dose COVID-19 vaccine[35]. The hollow sphere shape leads to a characteristic SAXS pattern with well-defined minima. Less characterized by SAXS, thyroglobulin is a 660 kDa dimeric protein produced in the thyroid gland. It is the main precursor to thyroid hormones, and thyroglobulin levels in blood are used as a tumor marker for thyroid cancer. The high-resolution spatial structure recently determined[36] shed light on the mechanism of hormone synthesis by thyroglobulin. Finally, the receptor-binding domain (RBD) from the SARS-CoV-2 spike protein was also measured. The RBD binds to angiotensin-converting enzyme 2 (ACE2) located on the surface of the cell membrane during the first steps of viral attachment. Once the virus is attached, it can then enter the cell to continue the replication cycle. The spike protein RBD is a primary target for drug development to prevent and treat viral infection. For example, therapeutic neutralizing antibodies can be designed to bind to the RBD and prevent its interaction with ACE2. Synchrotron SAXS was recently used to screen different synthetic nanobodies that bind to RBD and to propose a low-resolution structure of the complex formed between the RBD and a neutralizing sybody[37]. Capitalizing on the intense X-ray pulses, XFEL-SAXS could be a powerful tool for screening studies requiring many measurements, often at very low protein concentrations.

We show here that the scattering form factor of biological molecules can be accurately measured on XFEL instruments. To assess the quality of the collected data, control measurements were performed on well-characterized standard proteins. This ensured the quantification of experimental artifacts and validation of each step in data collection and treatment, including calibration, subtraction, analysis, and interpretation of the results. Obtaining form factors of biological macromolecules in solution using an XFEL source opens new scientific opportunities. Exploiting the X-ray pulse pattern, time-resolved experiments can be envisaged with unmatched time resolution. Difference scattering curves have thus far been collected and employed to monitor fast reactions. However, extraction of the scattering form factor affords less reliance on a priori information and hypotheses to analyze and interpret the data. This is particularly interesting for molecular reactions that result in changes of oligomeric state or complex formation. Collection of the matching buffer could be envisaged in addition to the difference patterns to combine the benefits of both approaches. These reactions are difficult to interpret using difference curves and would benefit greatly from establishing optimized XFEL-SAS data collection protocols. In addition, high-quality data collection from very dilute solutions of biomacromolecules is also made possible at an XFEL instrument optimized for solution scattering measurement. The highly intense beam and windowless environment open the technique to samples that can only be obtained in very limited quantities or that cannot be concentrated to the levels yielding meaningful signals at synchrotron sources.

## Results

**Reservoir delivery**. Samples and buffers were first delivered under pressure using stainless steel liquid reservoirs. The solution in the reservoir (sample or buffer) is continuously jetted through the X-ray beam, and data is acquired for each measurement. In this modality, separate solvent/buffer measurements and sample/protein measurements are necessary, requiring that several independent reservoirs are connected and filled with a sufficient volume of the respective sample or buffer.

All samples were prepared in the same buffer, and buffer curves were recorded regularly during the data collection. Here, a collection of different buffer curves shows variation in the scattering intensities, presumably due to variation or drift in the beam intensity and/or in the beam or jet position. To limit the impact of long-term variations, the buffer curves collected immediately before and/or after the sample were used for buffer subtraction. Still, some SAXS curves collected on the buffers had to be rescaled (up to 5%) before subtraction from the sample curves.

The form factors obtained via XFEL-SAS from BSA, apoferritin, and thyroglobulin are shown in Fig. 1. BSA and apoferritin, well-characterized proteins often measured by SAXS, serve here as standards to assess the validity of XFEL SAXS data. The form factors of both proteins (Fig. 1a, b) are in good agreement with those collected by synchrotron radiation SAXS and deposited in the publicly available biological SAS database, SASBDB[38]. A small discrepancy is observed between the experimental data sets for BSA at the lowest angles ($s < 0.5 \, \mathrm{nm}^{-1}$). Although these differences may originate from the presence of aggregates or large oligomers in the sample, the discrepancy may also be related to stochastic changes in the instrument background. Variation in the scattering background between individual samples and buffer measurements affects the buffer correction procedure and the reliability of form-factor recovery. In contrast to BSA, the XFEL SAXS data collected on Apoferritin show a good agreement with synchrotron data across the entire small angle range (Fig. 1b). It is likely that a better match in the instrument background was achieved for this series of sample and buffer measurements, but also that the larger size of apoferritin results in a stronger scattering signal at low angles. Thus, this measurement series is expected to be less sensitive to a small background mismatch.

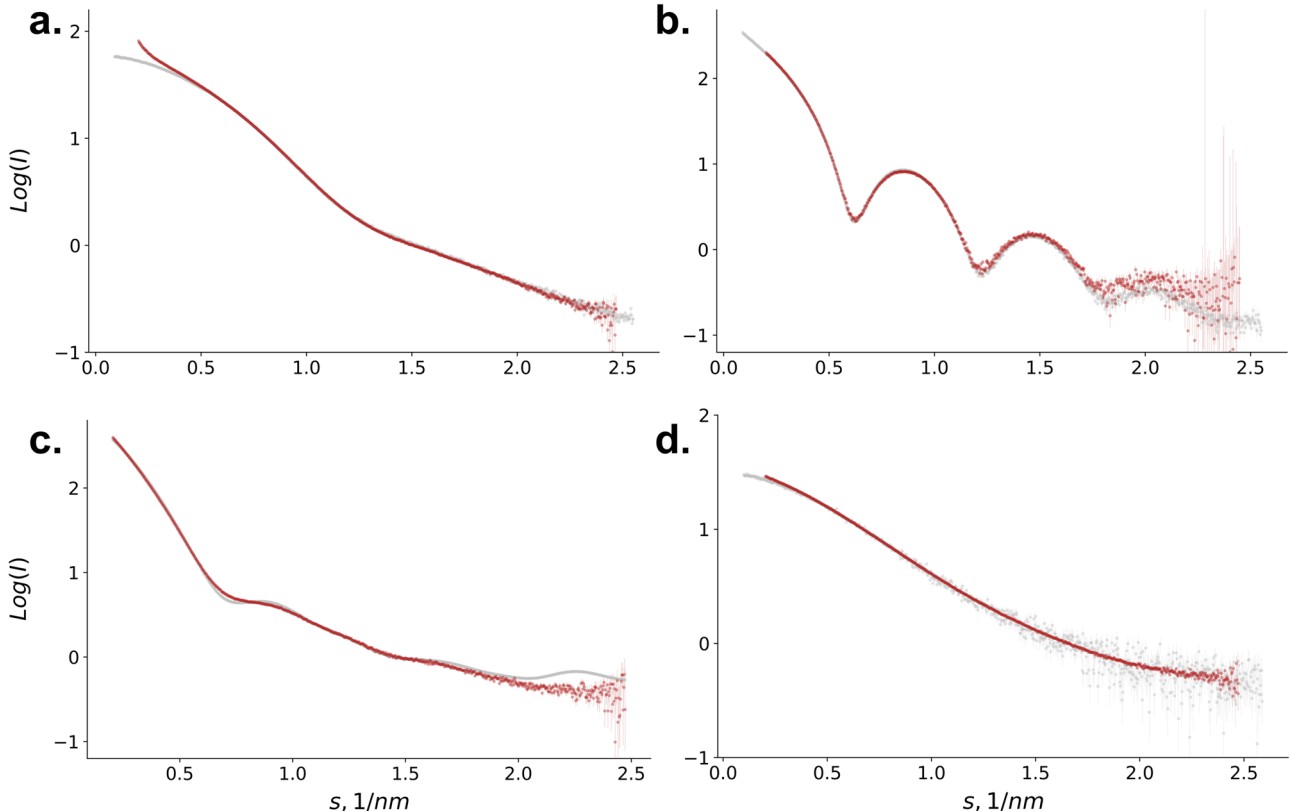

**Fig. 1 XFEL-SAS scattering form factors of protein samples.** Scattering form factors of Bovine serum albumin (**a**), apoferritin (**b**), thyroglobulin (**c**), and receptor-binding domain from SARS-CoV-2 spike protein (**d**). XFEL-SAS curves are in red. In gray, reference curves from the SASBDB database (BSA: SASDA32, Apoferritin: SASDA82, RBD: SASDJG4) or computed from atomic structure (thyroglobulin, pdb id: 6SCJ).

The form factor collected on thyroglobulin is reported in Fig. 1c. No crystal structure has been determined for thyroglobulin, and this has been attributed to conformational flexibility, and as such, thyroglobulin provides an interesting test case for scattering methods in solution. Recently two independent cryo-electron microscopy models have been determined for the human form, along with a model of the bovine thyroglobulin. The theoretical profile computed from the high-resolution cryoEM model of the bovine thyroglobulin dimer provides a reasonable fit to the experimental SAXS data and confirms the dimeric nature of thyroglobulin in solution. The coordinates of the C- and N-terminal residues, as well as residues in two flexible loops, are missing in the high-resolution structure. They have been modeled using the SAXS data acquired from the XFEL (Fig. S1). A rigid body modeling procedure was employed to introduce appropriate glycosylation and address the flexibility in the N-terminal domain suggested by Coscia[36]. A notable improvement in the agreement of the scattering from the refined models with the XFEL-SAS data was observed, and the functional impact of such conformational changes in this essential hormonal regulator is now a focus of further investigation.

The RBD form factor (Fig. 1d) is in good agreement with the SAXS curves available in SASBDB. One can note that the curves collected on XFEL show much lower noise than those from the database. This is only in part due to the higher concentration used for XFEL experiments (4.6 mg/ml vs. 1.1 mg/ml for synchrotron data). The good signal-to-noise ratio of the RBD curve underlines the potential of XFEL for BioSAXS data collection. The intensities at low angles in the Guinier region are slightly higher in the curve obtained on XFEL compared to the SAXS data. This can be explained by the presence of larger oligomers in the solution (as for BSA); the samples measured on XFEL have a higher

concentration than those measured on the synchrotron. The effect of background mismatch can also not be fully excluded.

To obtain the RBD form factor, no manual scaling of the buffer was required, and the average of the buffer collected just before and after the sample was used for subtraction. Although the SAXS intensities vary during collection—about 7% change between the buffer collected before and after the sample—the successive collection of buffer–sample–buffer allows one to interpolate the correct buffer for subtraction. This strategy can be employed to deal with slow-intensity drift, and it was implemented on first and second-generation synchrotrons (functioning without top-up mode) to collect BioSAXS data. This approach is still the default measurement scheme for many BioSAXS synchrotron beamlines and can conveniently be done using auto sample delivery presented below.

**Autosampler delivery**. To limit the impact of potential drift in the beam and/or jet position and reduce the sample volume, an autosampler was used. The experimental setup and principle of an autosampler are illustrated in Fig. 2. Here, the samples are directly injected into the buffer flow. The sample is moved with the buffer through the tube and into the jet. The dilution along the line is limited (Fig. S2), and the sample concentration remains high when it is jetted in the beam. Data is continuously acquired such that the curves can be successively collected on both the sample and the buffer (before and after the sample). The serial measurements of the buffer–sample–buffer in a single run are key to mitigating the effect of beam position and/or jet drift. The buffer measured before and after the sample can also be compared to verify that there is no cross-contamination along the delivery line (The autosampler typically has less than 0.005% of cross-contamination). The autosampler also allows for precise

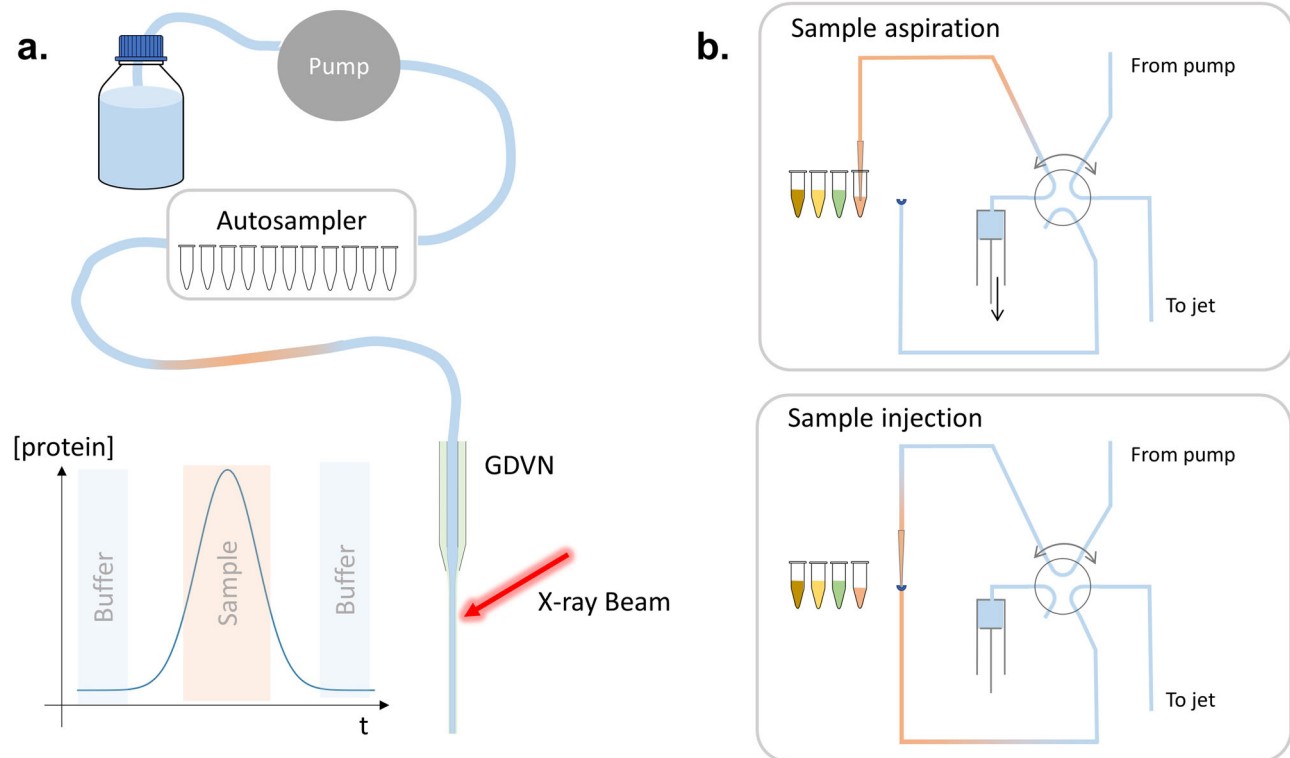

**Fig. 2 Autosampler operations. a** Sample delivery line with autosampler, the curves illustrate the protein concentration evolution in the jet. **b** Principles of in-flow sample injection used in the autosampler.

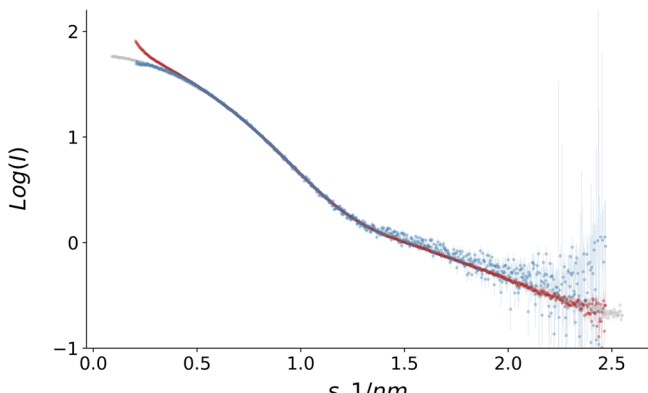

**Fig. 3 BSA form factor: reservoir vs. autosampler delivery.** BSA form factor collected on XFEL using reservoir (red) and autosampler (blue) delivery. In gray, synchrotron data from SASBDB (SASDA32).

control of the injected sample volume and for the possibility of serializing sample injection and data collection.

Figure 3 shows the data collected on BSA using autosampler and reservoir delivery overlaid with the SAXS curve from the SASBDB. The curves collected with the autosampler are in good agreement with the synchrotron data, especially at low angles. The higher signal-to-noise ratio in the autosampler can be explained by different factors. With the autosampler, 100 μl of sample volume was injected (750 μl with the reservoir), a reduced number of frames was selected for averaging (specifically those with the highest protein concentration, as shown in Fig. 2), and the concentration is slightly lower due to dilution along the line (Fig. S2).

The data collected on BSA illustrates well the value of the autosampler. The upturn at low angles observed with reservoir delivery disappears. Since the same sample was used in the

reservoir and autosampler, the discrepancy in the data is not the result of aggregates in the prepared solution. Aggregation in the reservoir between the filling and measurement cannot be entirely ruled out. Although the reservoir is cooled at 4 °C while mounted during measurement, there is no active temperature control available during filling and transport procedures. The protein concentration in the beam is higher when the reservoir injection method is employed compared to when the autosampler is used. With the autosampler, the protein sample is injected into the buffer flow, leading to dilution during transportation to the jet nozzle (see Fig. S2). The higher concentration in the reservoir sample delivery could potentially result in the formation of aggregates, causing an upturn at a low angle in the scattering curve. However, this is unlikely since BSA exhibits repulsive interactions at high concentrations and under similar buffer conditions[39]. The reason for the intensity upturn observed at low angles in the reservoir data is likely due to improper background subtraction due to the longer time gap between the buffer and sample measurements. Using the autosampler, the sample and buffer can be measured in the same run, seconds apart from each other. This approach reduces the time for the beam or jet to drift and thereby minimizes its effect on the resulting data.

## Discussion
Scattering form factor from protein in solution could be collected using X-ray produced by free electron lasers and used for modeling (see ab initio model in Fig. S3). However, extracting the extremely weak scattering signal of biological molecules from the scattering by the buffer and the instrument remains challenging for FEL sources due to the additional variations in the sample and X-ray delivery, which have to be excluded from obtaining reliable data from the samples. Below we discuss the current achievements, limitations, and possible ways to further improve BioSAXS on FEL.

An important challenge lies in keeping the scattering background low and stable within the timescales required to accurately measure the background. Figure S4 illustrates the long-term variations in the background, showing several buffer curves collected during the beamtime. These curves were obtained with different beam intensities as the sample reservoir was exchanged, and jet parameters were adjusted between measurements. The observed high disparity among the curves can be attributed to changes in the beam intensity or jet diameter. Interestingly, the curves can be overlapped once corrected using a scaling factor and offset. Ideally, the scaling factor and offset could be determined by measuring the beam intensity and the jet diameter, as well as their intersection with the beam. Nevertheless, achieving precise measurements for scaling and offset can be challenging. To address these challenges, the use of an autosampler becomes advantageous. The autosampler enables rapid data collection on both the sample and the buffer within a few seconds, reducing drift in beam intensity and jet geometry. By minimizing the time between buffer and sample measurements, the autosampler improves the reliability of background subtraction.

Variations at a short timescale, within a pulse train where each pulse induces an explosion of the jet, must be managed. Debris from the explosion of previous pulses may remain or pass through the beam trajectory interacting with subsequent pulses, resulting in a short-term increase in the background and thus spoiling the SAXS data. This effect was observed with the 1.1 MHz repetition rate and high transmission. It could be mitigated by reducing the repetition rate to 0.5 MHz (allowing the debris from the previous jet explosion to leave the interaction region) and by reducing the X-ray pulse intensity with the instrument attenuators (thereby lowering the amount of debris created by each pulse). Furthermore, each individual image collected within the train can be compared to the identified droplet scattering. In our experiments, the frames containing additional flares and background features associated with droplet scattering in the 2D images are filtered out prior to azimuthal integration (Fig. S6). Only the remaining non-affected images are used for further processing.

Changes in the scattering background in the 2D images were also observed due to variations in the jet position during measurement. Rapid fluctuations, flickers, and sputtering of the jet were minimized with careful control of the liquid and gas injection parameters. However, they cannot be eliminated completely, and features related to jet motions were also filtered during the 2D data reduction prior to radial integration. Longer-term jet movements are monitored with online visualization and corrected by readjusting the nozzle positioning motors. Large flares in the 2D images caused by the interaction of the X-rays with the edge of the jet were masked out of the 2D images and excluded from the radial integration. Changes in the scattering intensities coming from different path lengths (jet thickness) could not be corrected directly as there was no pulse-resolved intensity monitor available downstream of the sample. In practice, the intensity fluctuations over many pulses were averaged, and the scaling was corrected during buffer subtraction. Future implementation of a downstream intensity monitor would help to correct this effect but is not yet implemented at the SPB/SFX instrument.

Each pulse has variations in its intensity and the photon wavelengths contained. However, these variations are greater from train to train due to the intra-train feedback maintaining stability. The (shot to shot)/(train to train) intensity variations can be scaled using the measured pulse intensity from the upstream (XGM)[40]. This procedure enables scaling over time to correct for intensity changes between the measurement of the buffer and sample. Variations in the wavelength of the scattering, though small, are also present. Since there is no monochromator at the SPB instrument, some smearing of features related to the bandwidth of the SASE source (0.1%) is expected. However, this effect is minor in comparison to other possible experimental artifacts, and SAXS can tolerate minor variations in wavelength (note that pink beam and multilayer monochromator SAXS beamlines are common at synchrotron sources[41]).

Longer-term drift of the beam pointing is primarily attributed to the distribution mirrors. The drift was corrected with piezo control of the angle of the final distribution mirror to return the beam trajectory to the reference (as measured on the final imaging screen inside the SPB/SFX intelligent beam stop (IBS)[42]).

The influence of the drift can be mitigated by collecting buffers immediately before and after the sample. The contribution of the buffer to the sample scattering curve can be approximated by the average of the buffer collected before and after the sample. In this context, the autosampler was highly beneficial, allowing the injection of sample volume directly into the buffer flow. As such, one can collect buffers and samples in a single run within a few minutes. The autosampler also allows better control of the sample volume delivered, automation of the sample delivery, and serialization of the measurement. Furthermore, this system facilitates the inclusion of size-exclusion chromatography (SEC) columns for the inline purification of samples in the future.

The experiments can still be further improved by adjusting the instrument and sample injection, allowing us to fully exploit the exceptional characteristics of the FEL-X-ray beam. Utilizing FEL SAXS has the potential to greatly enhance Biological SAXS. The advantage of a windowless sample environment is that it reduces instrument background by eliminating any scattering caused by windows or sample containers. This, in turn, increases the sensitivity of the instrument. Jet sample delivery for SAXS necessitates precise control of a small beam at both the sample position (to match the jet size) and the detector position (to collect photons scattered at low angles). The coherent beam of XFEL uniquely facilitates achieving this control and represents an important advantage (It is worth noting that this concept could also be explored for modern beamlines, particularly those on 4th generation synchrotrons). The abundance of photons delivered by a free electron laser provides a competitive edge. In total, 50 µl of samples can be effectively illuminated by $10^{16}$–$10^{17}$ X-fel photons. This represents an increase of three to four orders of magnitude compared to the typical photon count employed to illuminate a 50 µl sample in synchrotron SAXS. Another advantage of FEL X-rays is that they are delivered in very short bunches. Synchrotron BioSAXS is limited by radiation damage, where free radicals produced by water radiolysis damage proteins, leading to aggregation and rapid degradation of the SAXS signal. By utilizing ultra-short pulses, all X-rays can be scattered before protein aggregation occurs. With an adjusted liquid jet, each subsequent X-ray pulse interacts with a fresh sample, thereby eliminating the issue of radiation damage. Implementing this approach in synchrotrons is unfeasible due to the longer and more frequent X-ray bunches. Quantifying the exact improvement of XFEL-SAXS for low-concentration samples presents challenges, and direct experimental comparisons with well-established synchrotron-based BioSAXS may not be fair at this early stage of XFEL-based studies. Nevertheless, the exceptional characteristics of the FEL-X-ray beam and the advantages it offers for sample delivery and radiation damage control suggest that XFEL-SAXS holds promise as a powerful tool for investigating challenging systems with low-concentration samples.

The high repetition rate of EuXFEL is beneficial to the amount of data that can be collected, taking advantage of the 2D filtering. However, the methods for form factor recovery described here could also be applied at lower repetition rate FEL sources. The degree of contamination of debris from previous pulses could be

lower, but as debris can still persist in the catcher, the need for removal of outliers cannot be eliminated completely. Obtaining form factors from data collections at lower repetition rate sources using lower jet speeds would enable greater access to FEL-SAS for the wider user community.

FEL pulse length also makes it the X-ray source of choice for ultra-fast time-resolved experiments. The collection of difference scattering curves (computing the difference between the data collected, for example, just before and after illumination) is already employed at XFEL and is well-adapted for reactions that do not involve any change in the oligomeric state. When protein dissociation or association is studied, the difference curve becomes difficult or even impossible to interpret. The method presented here would allow one to collect the full form factor, thus allowing one to study association/dissociation reactions. More generally, the autosampler allows precise control of the sample injection, which could also be beneficial for different experiments (for example alternating "on/illuminated" and "off/dark" frames while the sample elutes in the beam).

In summary, we report that form factors of proteins in solution have been successfully extracted from XFEL SAXS data. This procedure, summarized in Fig. 4, was facilitated by the incorporation of automatic sample injection into the flowing buffer using HPLC pumps which are used for the GDVN liquid jet injection. The high repetition rate yields important benefits enabling the filtering of outliers and rejecting frames containing variation in scattering flares from the edge of the jet and of the debris. However, the methods for form factor recovery described here could also be applied at lower repetition rate FEL sources. The scattering form factors measured on different macromolecules compare well with data collected on a synchrotron beamline and are perfectly interpretable. This is particularly well illustrated with the thyroglobulin FEL-SAXS data that were used here to model the fragments of the protein and also glycans missing in the available high-resolution structure. By obtaining the proof of principle of recovering the form factor of proteins from the solution scattering and intrinsic background fluctuations of pulsed SASE source, this work enables future projects of greater biological relevance to be considered. Scope for investigations at shorter time scales and at potentially lower sample concentrations can be considered as part of future work to optimize the experimental parameters providing yet more efficient use of experiments to exploit FEL scattering with biological and biomedical samples in solution.

## Methods

**Instrument setup**. The SPB/SFX instrument[42] is a highly flexible endstation predominantly focused on life science experiments. SPB/SFX provides a configurable sample-to-detector distance and optimizable energy range coupled with micron focusing and background optimization for serial crystallography, single photon-counting single-particle diffraction experiments, and SAXS.

Although the SPB/SFX instrument can operate continuously between 6 and 15 keV, in practice, the use of optimized set-points delivers particular benefits. This optimization improves the stability of beam delivery and alignment, an important consideration for reliable background subtraction. Accordingly, the routinely used energy set-point at 9.3 keV was chosen for this experiment. With the aim of maximizing the number of images collected, operation at 1.1 MHz with 352 pulses per (10 Hz) train (3520 pulses per second) was planned, as well as 0.5 MHz with 202 pulses per (10 Hz) train to compare data quality. The maximum number of pulses that may be measured per train is limited by the available detector technology[43], which is unique for this high repetition rate facility.

To closely match the size of the liquid jet (Ø5 μm, calculated), the X-ray beam was focused using the KB mirrors at the upstream interaction region and interaction plane to a spot size of 4.4 μm (h) × 6.3 μm (v). The X-ray scattering background was minimized using the instrument slits in accordance with standard practice for solution SAXS experiments[16]. A distance of 3 m from the sample jet to the AGIPD1M detector[43] was chosen to ensure the Guinier range should be visible (>30 points) for all samples and to maximize $s$ range and the wider angles, to maximize the visible s range (0.057–2.46 nm$^{-1}$) with the available detector geometry (beam passing through the center of the detector). The position of the detector and its modules was adjusted to bring the beam position close to the active area in the vertical direction (Fig. S5). Calibration of the measured scattering angles was performed using observed scattering rings from silver behenate, the intensity is in arbitrary units.

**Sample preparation**. Samples were prepared according to standardized approaches for synchrotron SAXS data collection[44].

The RBD from Spike glycoprotein, SARS-CoV-2, was expressed in HEK293-F cells and purified as previously described[37]. Purified RBD was stored frozen (−80 °C) and dialyzed overnight into a PBS buffer before measurement.

Standard proteins were purchased from Sigma (Darmstadt, Germany): Bovine Thyroglobulin (Sigma-T1001), Equine Apoferritin (Sigma-A3660), and Bovine Serum Albumin (Sigma-A7030). Each standard was dissolved in a PBS buffer and filtered using an Ultrafree®-MC spin-filter device (12,000×$g$, 4 min.) prior to measurement.

**Sample delivery**. For reservoir delivery, samples were delivered at the SPB/SFX instrument via gas dynamic virtual nozzles (GDVN) using pressurized sample reservoirs[45]. The sample line was pressurized with Shimadzu HPLC pumps (LC-20AD), and sample flow rates were 50 μl/min. The following protein concentrations were used (BSA: 10 mg/ml, Apoferritin: 2.5 mg/ml, Spike RBD: 4.6 mg/ml, Thyroglobulin: 0.7 mg/ml). Enabling data collection with a 1.1 MHz or 0.5 MHz repetition rate, the helium mass flow was adjusted to reach jet velocities above 40 m/s or 25 m/s, respectively. GDVNs were 3D printed (Photonic Professional GT, Nanoscribe) with a liquid orifice of 75 μm and a gas orifice of 60 μm.

For Autoloader delivery, replicating the standard methods used at dedicated BioSAXS beamlines at synchrotrons[13,46,47] samples (50–500 μl) were placed in 1.5 ml Eppendorf microcentrifuge tubes and positioned in a temperature-controlled SIL-20AC autosampler (Shimadzu). Samples were injected via the Shimadzu HPLC pump system directly connected to a GDVN or double-flow focusing nozzle (DFFN)[48] using the Shimadzu control software remotely operated over the network. The injection volume was 100 μL for each sample; protein concentration was in the range from 0.2 to 8.5 mg/ml. To reach jet velocities above 25 m/s (for 0.5 MHz repetition rate), the sample flow was 50 μl/min, and the helium mass flow was 16 mg/min for GDVNs (3D printed; 75 μm liquid orifice, 60 μm gas orifice). For DFFN (3D printed; liquid orifice 100 μm, gas orifice 90 μm), the sample flow was 25 μl/min, the Ethanol flow 15 μl/min, and the Helium mass flow 34 mg/min.

**Online data monitoring**. Online data processing, monitoring, and analysis were performed using components of the SCADA ecosystem Karabo[49] and the EXtra-foam tool[50]. In Karabo, AGIPD1M detector data was streamed from the monitoring output channel of the data acquisition system to the processing pipeline for multi-gain pixel intensity correction from calibration

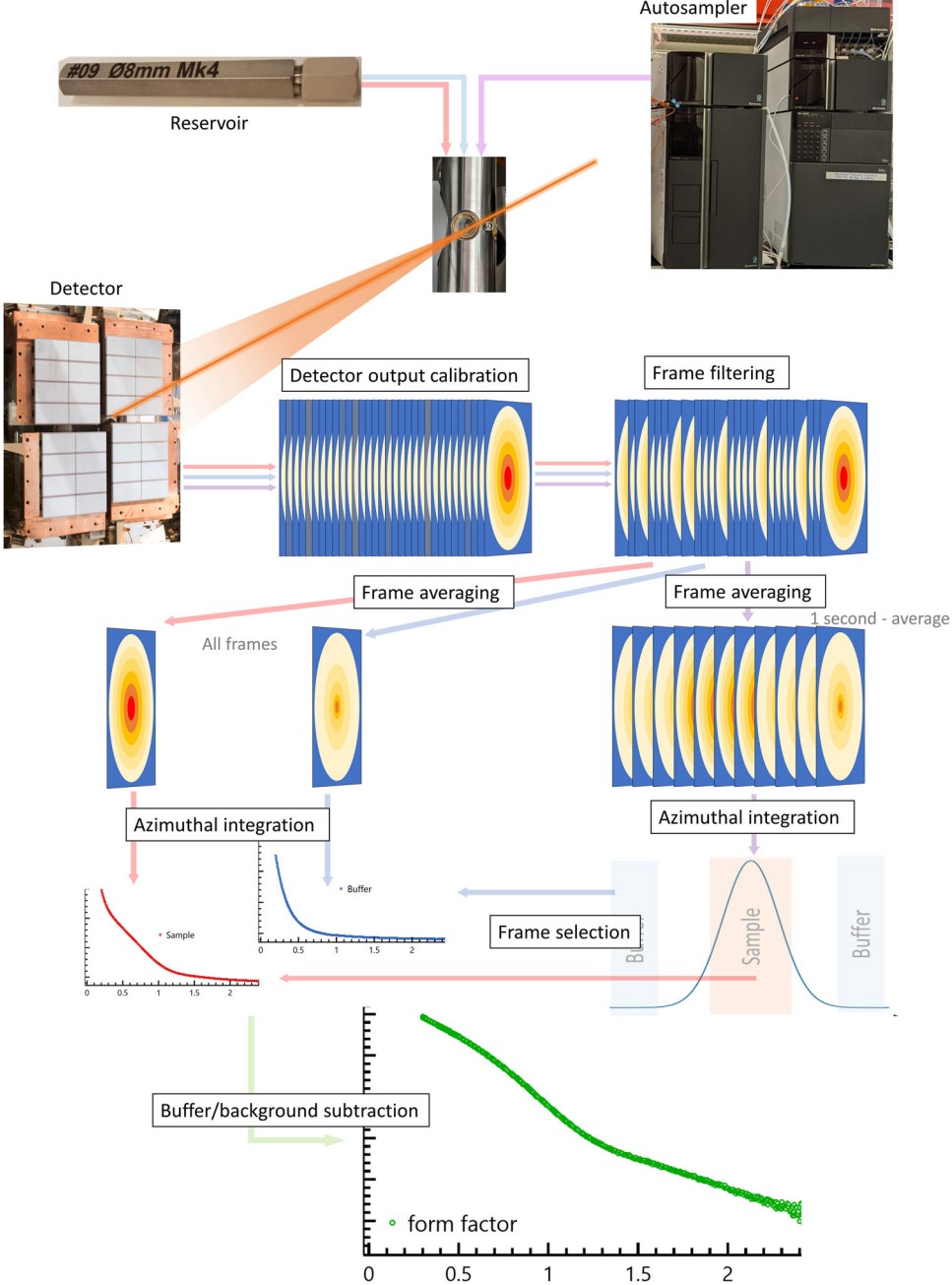

**Fig. 4 XFEL data collection and processing workflow.** Samples are injected using either the reservoir or the autosampler. Scattered X-rays are collected on the detector and processed. In the case of reservoir delivery, all filtered images are averaged and then azimuthally averaged. For autosampler data, the filtered data are averaged in 1-s intervals and azimuthally averaged. The resulting scattering intensity is used to plot the elution trace and select the sample and buffer frames. The scattering signal from the pure buffer is subtracted from the sample scattering to obtain the protein form factor.

constants ("online calibration pipeline") and monitored using the standard detector image previews (maximum intensity image from a train) with 1 Hz refresh rate in the Karabo GUI. The online-corrected data were further streamed to the stand-alone GUI application EXtra-foam to display the geometrically assembled multi-module detector images and calculate the scattering curves I(s) by first averaging pixel intensities from all detector image frames belonging to the same pulse train and then integrating the averaged intensities over the full azimuthal coordinate range for each radial distance from the detector center. For background subtraction, a feature of EXtra-foam to record an averaged reference background image from a number of sample-free frames stretching over multiple trains was used, followed by subtraction of re-loaded background runs from the live data with sample scattering.

**Offline data processing**. The data processing was performed in two stages, first, to calculate appropriate pixel-wise averages from the AGIPD detector, followed by azimuthal integration and small-angle scattering analysis. The first step began with the standard European XFEL detector calibration output. This consists of various stages comprising dark offset subtraction, baseline shift correction, and pixel-wise relative gain correction. The gain switching capabilities of the AGIPD were not relevant for the signal levels in this experiment. These calibrated values were

converted to photons using a threshold of 0.7 photons, where the number of ADU (analog-to-digital unit) counts per photon was estimated by examining the histogram of ADU values and identifying the 1-photon peak at 69 ADUs.

Filtering of "bad," i.e., inconsistent shots: Due to fluctuations of the liquid jet, not all frames were suitable for inclusion into the average pattern (see Fig. S6). This selection was done with a frame-wise metric consisting of 4 numbers representing the number of pixels with at least one photon in each of the inner 4 modules of the AGIPD. This 4-vector is a measure of scattering from the edge of the liquid jet or from droplets in the jet breakup. Additionally, since the beam center was not symmetrical relative to the position of the modules, this metric was also sensitive to the relative strength of such undesirable coherent effects compared to the small-angle scattering from the buffer or sample buffer liquid at higher scattering angles. The DBSCAN algorithm[51] was used to detect outliers using this frame-wise metric for each AGIPD cell separately, rejecting around 1–2% of frames depending on the run.

Frame averaging: After selection, the photon-converted frames were averaged depending on the type of data collection. For the autoloader runs with SEC-type injection, 1-s averages were performed, consisting of at most 2010 frames, to enable monitoring of the change in scattering varying with time. For the reservoir runs, the whole ~5-min run was averaged, as conditions were expected to be stable over the whole length of the run. This was checked by making additional averages in runs and comparing. Output averages (the whole train for reservoirs or stack of 1-s averages for the autoloader runs) were written to HDF5 files. Each average was converted to two integer TIFF files in order to interface with the ATSAS pipeline described below, namely the *sum* of the number of photons per pixel and the *count* of how many frames contributed for each pixel. Since the set of bad pixels of the AGIPD is different for each cell, this latter image had small variations from pixel to pixel. All calculations were performed with custom *Python* scripts using the multiprocessing and mpi4py libraries for parallelization on the SLURM batch system of the Maxwell computing cluster at DESY/EuXFEL.

Azimuthal integration: Azimuthal integration of the detector image files (*.tif) was performed using components of the SASFLOW pipeline from the ATSAS software package[52,53], yielding 1D SAXS profiles I(s) vs. s, where s = $4\pi \sin\theta/\lambda$, and $2\theta$ is the scattering angle at a wavelength $\lambda = 0.13$ nm (and energy 9.3 keV). For samples delivered using gas-pressurized reservoirs, all remaining frames post filtering were averaged for each run. For buffer subtraction, buffer frames collected directly before and/or after the sample frames were used to limit the impact of long-term drift. Additional scaling of buffer scattering intensities was conducted as required (up to 5%) to obtain accurate scattering data at the higher angles. Data analysis and buffer subtraction were performed in PRIMUS/Qt[54]. For samples delivered via the autoloader system, stacks of 1 s frame averages were analyzed by CHROMIXS[55], and buffer/sample identification and subtraction were performed interactively.

**SAXS data analysis and modeling**. All SAXS data were analyzed using PRIMUS/Qt[54] and the ATSAS software package[53]. The forward scattering I(0) and radius of gyration, $R_g$ was determined from Guinier analysis[56] based on the assumption that at very small angles ($s \leq 1.3/R_g$), the intensity is represented as $I(s) = I(0) \exp(-(sR_g)^2/3)$. These parameters were also estimated from the full scattering curves along with the distance distribution function $p(r)$ and the maximum particle dimensions $D_{max}$, using the indirect Fourier transform method implemented in the software GNOM[57]. Computation of theoretical scattering intensities was performed using the program CRYSOL[58].

Ab initio structure calculation—low-resolution structures representing the molecular form factor were reconstructed from SAXS data using the programs DAMMIF and DAMMIN[59,60], which represent the macromolecule as a densely packed interconnected configuration of beads that best fits the experimental data $I_{exp}(s)$ by minimizing the discrepancy:

$$\chi^2 = \frac{1}{N-1}\sum_{j=1}^{N}\left[\frac{I_{exp}(s_j) - cI_{calc}(s_j)}{\sigma(s_j)}\right]^2 \qquad (1)$$

where $N$ is the number of experimental points, $c$ is a scaling factor, and $I_{calc}(s_j)$ and $\sigma(s_j)$ are the calculated intensity and the experimental error at the momentum transfer $s_j$, respectively. The stability of the DAMMIF model solution was verified through multiple independent modeling runs, and a consensus model was obtained through refinement of the search volume in DAMMIN.

Hybrid rigid body modeling—rigid body models representing the molecular form factor were computed from the experimental SAXS data using the software CORAL[61]. The cryoEM structure of Thyroglobulin (PDB ID. 6SCJ) was used to define the rigid bodies, with residues missing in the cryoEM structure added (24 at N-terminal and 40 at C-terminal and flexible linker residues defined between residues 1513-1564 and 1954-1962). Appropriate glycosylation (16 ASN-linked sites) was introduced into the models using the GLYCOSYLATION routine of ATSAS.

Normal modes model refinement—for cases in which the theoretical scattering computed from a high-resolution structure was in poor agreement with the experimental SAXS data, SREFLEX[8] was used to refine the high-resolution model utilizing the transformations along the calculated normal modes to improve the fit to the experimental data.

**Statistics and reproducibility**. Data processing involved the use of custom Python scripts, thoughtfully designed with parallelization techniques, to handle various critical tasks. These included detector calibration, filtering out inconsistent data frames, and performing frame averaging.

For form factor analysis, we relied on the well-established and trusted methods available within the ATSAS package. These widely recognized techniques includes azimuthal averaging, guinier analysis, inverse fourier transform, computation of scattering curves from atomic model, ab initio and hybrid modeling.

**Reporting summary**. Further information on research design is available in the Nature Portfolio Reporting Summary linked to this article.

## Data availability

SAXS-FEL data have been deposited at the SASBDB (www.sasbdb.org) with accession codes: SASDQX8, SASDQY8, SASDQZ8, SASDQ29, and SASDQ39. All other data are available from the authors on reasonable request.

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

## Acknowledgements

We acknowledge European XFEL in Schenefeld, Germany, for provision of X-ray free-electron laser beamtime at Scientific Instrument SPB/SFX (Single Particles, Clusters, and Biomolecules and Serial Femtosecond Crystallography) and would like to thank the staff for their assistance. The authors acknowledge the in-kind support of the DESY Strategy Fund Corona-related research project scheme (200702, to C.S., H.C., K.A., C.M.J. and D.S.V.). This work is also partly supported by the Cluster of Excellence "CUI: Advanced Imaging of Matter" of the Deutsche Forschungsgemeinschaft (DFG)-EXC 2056-project ID390715994.

## Author contributions

C.B. and A.R.—Experimental design and planning, data collection, data analysis and writing the paper. H.M.— Sample preparation, data collection, data analysis, and writing the paper. K.A., Ab.M., T.W., Y.Z.—2D data reduction and analysis. M.G.—Experimental design and planning, sample preparation, delivery and data collection. C.J.—Sample preparation, delivery and data collection. K.D., M.K., Ju.K., S.A., G.E.P.M., A.H., D.O., S.B. and J.S.—Sample delivery and data collection. D.F. and A.G.—Data reduction and processing from 2D and 1D. J.V., E.R. and J.M.—Sample preparation and delivery. M.S., R.L., R.B., R.W., J.E., Y.K., H.K., Ja.K., P.V.—Data collection. R.M., T.C. and C.L.—Sample preparation. H.C., Ad.M. and D.S.—Experimental design and planning, paper preparation. H.H.—Sample and paper preparation. C.S.—Paper preparation. All—Paper approval.

## Funding

## Competing interests

The authors declare no competing interests.
