## [Peer Review File · Communications Biology]

Reviewers' comments:

Reviewer #1 (Remarks to the Author):

Blanchet et al. conducted a study that determined the form-factor of biological molecules (protein-only SAXS signal) using X-ray free-electron laser (XFEL) sources. The authors measured scattering signals for proteins of various sizes in solution, as well as the background signals contributed by the solvent and XFEL instruments. From the scattering signals, the authors were able to extract the scattering signals for pure proteins by subtracting the background signals from the protein solution signals, as typically done in SAXS experiments at synchrotrons. Notably, the authors demonstrated that using the autosampler delivery system during SAXS measurements at XFELs can effectively reduce the intrinsic background fluctuations caused by the XFEL instruments, allowing for a relatively accurate extraction of the scattering signal for pure proteins.

The authors mentioned the necessity and novelty of this study as follows. Although BioSAXS has been widely performed at synchrotrons, it was not easy to apply for proteins that are difficult to purify due to the requirement of relatively high concentrations. Accordingly, the authors aimed to perform BioSAXS experiments on proteins at low concentrations using an XFEL, which provides much intense X-ray pulses. The authors also stated that X-ray solution scattering studies on proteins, including time-resolved experiments, have been conducted so far, but there has been no report on protein-only SAXS signals.

Nevertheless, regrettably, this reviewer encountered difficulties in identifying any novel analytical or experimental advancements in this manuscript. The procedures for obtaining protein-only SAXS signals, for example, are already well established from numerous synchrotron-based BioSAXS experiments. Additionally, sample delivery systems suitable for XFEL experiments have already been employed in previous X-ray solution scattering studies on proteins. Lastly, considering that more sophisticated time-resolved SAXS/WAXS data have already been published, the detection of static SAXS signals is not considered to have significant impact by this reviewer. Consequently, this manuscript cannot be recommended for publication in *Communications Biology*. Some other comments are listed in the following.

1. This reviewer suggests that it would be more beneficial for the manuscript to place greater emphasis on the methodology employed for collecting and processing SAXS data and the authors' contributions to enhancing these procedures, rather than solely focusing on the experimental SAXS data. Furthermore, it would be useful to discuss the advantages of utilizing XFELs over synchrotrons for SAXS data acquisition.

2. The concentration of the sample used in the autoloader delivery-based experiment was lower than that used in the reservoir delivery-based experiment. As such, it is recommended that the effect of this concentration difference on the BSA scattering signals be discussed in more detail.

Reviewer #2 (Remarks to the Author):

Comments and suggestions

The authors demonstrated that they could reliably measure the form factors of biological molecules in solution by using an XFEL source. It can open up new potential for the study of the structure and dynamics of biological molecules in solution.

In the introduction, the authors talk about important and relevant issues such as SAXS in solution vs crystallography and cryoEM, synchrotron vs XFEL source, sample concentration issue, systematic error issue of background scattering subtraction, and so on. They applied the XFEL-SAS method specifically to some interesting model proteins. The authors worked on standardization and automation delivery of samples to the X-ray beam and utilized the high repetition rate of XFEL which helped walk around the

inherent instability problem of the XFEL source. The conclusions are drawn reasonably based on their experimental results.

The manuscript reads well. I recommend publication in Nature Communications Biology after the authors address some questions/suggestions and minor corrections.

Here are some questions/suggestions.

1. On lines 164-167, the authors set their goal by mentioning "For direct form factor recovery from XFEL-SAS experiments complications arising from the variability of X-ray pulses and the stability of the liquid jet sample delivery must be overcome, and this forms the main goal of this work."

The manuscript would be more informative and appealing if some figures/data regarding the experiments complications are shown instead of just being stated in the text.

For example, it would be easier to understand the problems if the authors show the variation of buffer scattering due to X-ray beam (SASE) or sample jet instability.

2. In multiple places (lines 89, 101, 113, and 370 ...) authors mentioned XFEL-SAXS measurement of a dilute sample system. If the background scattering due to solvent/buffer (and/or something else) is the biggest issue in a dilute sample system, how is XFEL more beneficial over synchrotron source if the total number of photons matters at the end? It would be more convincing if authors elaborate better on the benefits of XFEL over synchrotron source to extract the form factor from a dilute system.

3. Figure 1(c) shows a bump-up at around 2.0-2.5 [1/nm] in the calculated scattering curve of thyroglobulin which is not shown in Figure S2. What are the differences between the two figures?

4. With flow rate 50uL/min (>25m/s) for 0.5MHz repetition experiment, an X-ray spot size of 4.4 um(h)x6.3 um(v) is small enough to replenish the sample between pulses.

It is helpful if authors specify X-ray power (or energy per pulse) and its consequent temperature-jump by X-ray absorption, including the initial temperature. It would be good to add the estimation or measurement of temperature.

5. Did the authors happen to make any effort in order to minimize the dilution/mixing by switching samples in the autosampler delivery system? How did the authors handle the contamination issue in the experiment and data analysis if there is any?

Minor corrections:

1. On line 270, figure 3 should be figure 2.

2. The x-axis scale in figure S2 is not consistent with figure 1(c).

3. The concentrations of RDB have inconsistencies in two places: 4mg/ml (line 252) vs 4.6mg/ml (line 420).

Reviewer #3 (Remarks to the Author):

This was a very clearly written manuscript describing proof-of-principle experiments attempting to extend Small Angle Scattering (SAXS) of biological samples, an established technique from the synchrotron community, into X-ray Free Electron Laser facilities. While the efforts are heroic and the authors had to overcome many technical challenges, which they describe in detail, the motivation for such experiments, especially non-time resolved, are a bit under-explored in my opinion. However, I believe that increasing the scope of techniques available at XFELs is a worthy pursuit and there is

potential in this method. I would recommend the manuscript for publication after revision.

My specific comments follow below:

1. As a non-SAXS expert, it is not clear to me exactly what information is directly encoded in the form factor that cannot be obtained by other methods such as SFX (already widely established at XFELs) and cryo-EM. It seems from the manuscript that these are complementary measurements for a particular set of systems that might be hard to obtain otherwise. Is that correct? If so, is the scope of such measurements wide enough to warrant the difficult experiments and specialized setups necessary for such experiments?
2. The authors cite as an advantage the intense and short pulses of XFELs. I understand the argument about the photons per pulse, but the time-resolution is not used at all in this study. Is the goal (in the long term) to extend the absolute form factor studies to the time-resolved domain? And if so, wouldn't relative changes be enough to determine photoinduced changes? Would it be really necessary to extract the absolute form factors of photoexcited systems? Why?
3. Since the low sample consumption needed for the XFEL experiments compared with the synchrotron studies is mentioned as another advantage, would it be possible to give an estimate of the reduction in sample consumption compared to traditional BioSAXS experiments? Isn't this advantage negated by the extremely high jet speeds needed to replenish the jet with the MHz repetition rates used in the experiment?
4. On the same note, the authors mention that the sample consumption can be further reduced with the autosampler. I don't follow how that is possible since the jet speed has to be the same. Is the argument here that less data needs to be collected to obtain curves with similar S/N? This is also my issue with Figure 3. The autosampler data has poorer S/N overall (and specially at high Q's) and the authors say that this is due to the "less sample being injected". Ideally one would show data collected with the same number of shots to demonstrate the advantages of the proposed method. I understand this might not be possible anymore due to the nature of beamtime, but one could easily plot the curves obtained by the traditional reservoir method scaled down to the same number of shots as that of the autosampler for a more direct comparison. If this is possible, I would very much like to see this figure in the SI
5. Another general comment: A lot of the technical challenges the authors report seem to be specific to high-repetition rate X-ray free electron lasers. I understand that is where they did the experiment but I would recommend that this shouldn't be generalised to all XFELs, since most of the hard X-ray FELs are actually operating in the 30-120 Hz range. Specifically, issues with inter-train intensity variations, jet explosion, fast jet operation and normalization are all non-existent or heavily mitigated when using a low repetition rate source. Have the authors considered performing the same experiments at a different facility? Either the authors should change the title/text to reflect the high-repetition rate XFEL nature of the challenges or comment on the expected performance at other XFEL facilities.

Response to referees

Reviewer #1:

1. This reviewer suggests that it would be more beneficial for the manuscript to place greater emphasis on the methodology employed for collecting and processing SAXS data and the authors' contributions to enhancing these procedures, rather than solely focusing on the experimental SAXS data. Furthermore, it would be useful to discuss the advantages of utilizing XFELs over synchrotrons for SAXS data acquisition.

We appreciate the reviewer's suggestion to emphasize the methodology and we have included supplementary figures (S6 and S7) that provide a more detailed illustration of the experimental workflow and data filtering steps.

Furthermore, we have expanded the discussion on the advantages of utilizing XFELs over synchrotrons for SAXS data acquisition (lines 372-386). XFELs offer significant benefits for time-resolved experiments due to their short, bright pulses. Additionally, they enable data collection before protein alteration occurs, mitigating issues related to radiation damage.

We believe that these additions enhance the manuscript by providing a more comprehensive understanding of our methodology and highlighting the advantages of XFELs in SAXS studies.

2. The concentration of the sample used in the autoloader delivery-based experiment was lower than that used in the reservoir delivery-based experiment. As such, it is recommended that the effect of this concentration difference on the BSA scattering signals be discussed in more detail.

We have provided a more detailed discussion (lines 304 - 310) on the effect of this concentration difference, accompanied by additional data presented in Figure S2. This data demonstrates how the sample is diluted along the delivery line to the jet nozzle.

Reviewer #2:

1. On lines 164-167, the authors set their goal by mentioning "For direct form factor recovery from XFEL-SAS experiments complications arising from the variability of X-ray pulses and the stability of the liquid jet sample delivery must be overcome, and this forms the main goal of this work."

The manuscript would be more informative and appealing if some figures/data regarding the experiments complications are shown instead of just being stated in the text.

For example, it would be easier to understand the problems if the authors show the variation of buffer scattering due to X-ray beam (SASE) or sample jet instability.

We have addressed the reviewer's comment by providing additional figures to illustrate the experimental challenges faced in XFEL-SAS experiments (see Figure S4). In this figure, we present different data sets collected on the buffer throughout the beamtime, highlighting variations in the scattering pattern attributed to drift, absence of beam intensity normalization, and differences in beam geometry. Interestingly, when the curves are scaled and adjusted by a constant offset to account for differences in beam intensity and jet geometry, they exhibit significant overlap, indicating the consistency of the data despite apparent differences.

Furthermore, we have included an example of accepted and rejected frames in the data processing pipeline (see Figure S7). The 'good' frames, which still show some scattering from the slits and/or jet edge, exhibit scattering primarily located at the vertical and horizontal positions of the direct beam, making it easily maskable. On the other hand, the 'bad' frames, likely resulting from jet instability or debris in the beam path, display stronger parasitic scattering and are appropriately filtered out.

2. In multiple places (lines 89, 101, 113, and 370 ...) authors mentioned XFEL-SAXS measurement of a dilute sample system. If the background scattering due to solvent/buffer (and/or something else) is the biggest issue in a dilute sample system, how is XFEL more beneficial over synchrotron source if the total number of photons matters at the end? It would be more convincing if authors elaborate better on the benefits of XFEL over synchrotron source to extract the form factor from a dilute system.

One major limitation for the measurement of a dilute sample system indeed lies in the reduction of the instrument background.

One significant advantage of using an XFEL with a liquid jet sample delivery is the elimination of the scattering background originating from the sample cell, typically made of capillary quartz. By using a liquid jet, we bypass the need for a solid sample holder, thereby reducing the background contribution.

While the total number of photons is a crucial factor, the structure of the XFEL X-ray pulse also plays a vital role. One major limitation of synchrotron beamlines is radiation damage to the protein sample. Radiation-induced free radicals lead to protein damage and subsequent aggregation, which rapidly deteriorates the scattering signal. In contrast, the ultra-short duration of XFEL pulses allows all X-rays within the pulse to scatter off the sample before the protein begins to aggregate. Moreover, when a properly adjusted liquid jet is used, the subsequent pulse interacts with a completely fresh sample. This approach effectively eliminates radiation damage-related issues that cannot be effectively addressed with synchrotron sources due to longer pulse durations and shorter time intervals between pulses.

We have added a new paragraph in the manuscript to provide a more detailed explanation of the potential benefits of using XFEL compared to synchrotron sources for BioSAXS experiments. (lines 370 - 384)

3. Figure 1(c) shows a bump-up at around 2.0-2.5 [1/nm] in the calculated scattering curve of thyroglobulin which is not shown in Figure S2. What are the differences between the two figures?

In figure 1C, the curves are directly computed from the pdb curves. In figure S2, missing linkers and glycans are added, and their conformation is adjusted against the SAXS data. This has been clarified in the paragraph reporting on the thyroglobulin scattering profile (lines 248 - 252) and in the legend of figure 1 and supplementary figure 1.

4. With flow rate 50uL/min (>25m/s) for 0.5MHz repetition experiment, an X-ray spot size of 4.4 um(h)x6.3 um(v) is small enough to replenish the sample between pulses. It is helpful if authors specify X-ray power (or energy per pulse) and its consequent temperature-jump by X-ray absorption, including the initial temperature. It would be good to add the estimation or measurement of temperature.

The temperature jump is sufficient to induce rapid conversion of the liquid to gas. This in turn pushes the surrounding liquid away from the interaction region. The Jet speed therefore has to be sufficient to be replenished prior to the arrival of the next pulse. With the correct combination of jet speed and repetition rate the scattering observed from the pulse later in the train are not statistically different from the first. The diffraction from the short pulses has been demonstrated to occur prior to the destruction of the sample and is used routinely for crystallography where the high resolution information obtained has been shown to be free from effects of the X-ray pulses.

5. Did the authors happen to make any effort in order to minimize the dilution/mixing by switching samples in the autosampler delivery system? How did the authors handle the contamination issue in the experiment and data analysis if there is any?

We took measures to minimize dilution and mixing effects during the sample transport in the autosampler delivery system. The extent of sample dilution along the transport line was tested prior to the experiment and is now presented in figure S2 of the manuscript. By using small diameter tubes, we were able to limit the dilution, ensuring that the sample concentration remained within the same order of magnitude.

To address the issue of contamination, we implemented a strategy to prevent cross-contamination between samples. Sufficient time intervals were incorporated between sample injections to allow for the collection of pure buffer before and after the sample elution peak. Cross contamination inside the autosampler is reported to be below 0.005%. We have now included these details in the revised manuscript (lines 274 - 281)

Minor corrections:

1. On line 270, figure 3 should be figure 2.

The figure number has been corrected

2. The x-axis scale in figure S2 is not consistent with figure 1(c).

The x-axis was in inversed angstroem and not inversed nanometer as indicated, this has been corrected (now figure S1).

3. The concentrations of RDB have inconsistencies in two places: 4mg/ml (line 252) vs 4.6mg/ml (line 420).

The concentration was 4.6 mg/ml and the 4mg/ml (line 256) was a typo and has been corrected (the RBD concentration of the synchrotron curve was also corrected, 1.1 mg/ml instead of 1 mg/ml as previously reported)

Comments in the file attached by reviewer 2:

- Lines 70-71: application of SAXS to lipid nanoparticles
 - Reference to relevant publications has been added (line 70)
- Lines 90-92, 100-11, 113-115, 132-136::
 - Related to comments 2
- Lines 116, 210-211:
 - Related to comment 1
- Lines 146-150, 158,,:
 - Related to comment 4
- Line 177:
 - Thank you!
- Line 181:
 - Corrected
- Lines 255-257:
 - This is also suggested by the comment 2 of reviewer1. We added a discussion (lines 300-309) and a supplementary figure (S2) to discuss difference in concentration in the reservoir and autosampler methods. The structure factor would however not be the likely explanation for the increase of signal at low angles, since in BSA is known to show repulsive interactions at high concentrations in these conditions. This would lead to a decrease in the scattering intensity at low angle for the most concentrated sample (delivered through the reservoir), and not an increase as observed here. This discussion and reference to studies of the intermolecular interaction of BSA at high concentrations have been added.
- Lines 293-296:
 - Linked to comments 1 and 2 + buffer now shown in figure S4
- Coments in the conclusion:

- Related to comment 1 + comment 4

Reviewer #3:

1. As a non-SAXS expert, it is not clear to me exactly what information is directly encoded in the form factor that cannot be obtained by other methods such as SFX (already widely established at XFELs) and cryo-EM. It seems from the manuscript that these are complementary measurements for a particular set of systems that might be hard to obtain otherwise. Is that correct? If so, is the scope of such measurements wide enough to warrant the difficult experiments and specialized setups necessary for such experiments?

In SAXS, biological molecules are studied in solution, eliminating the need for crystallization (as in crystallography) or freezing (as in Cryo-EM). By modifying buffer conditions such as salt concentration, pH, or introducing partner molecules, valuable information about induced structural changes can be rapidly obtained. SAXS is also highly useful for characterizing mixtures of different oligomeric states or flexible proteins that are challenging to study using other methods.

While the appreciation of BioSAXS methods may vary subjectively, it is important to recognize their broader application beyond being complementary measurements for specific systems. SAXS is a versatile technique that finds utility across a wide range of systems, including proteins, nucleic acids, lipid nanoparticles, and structures with nanoscale inhomogeneous electron density. BioSAXS is particularly useful for studying polydisperse samples, such as oligomeric mixtures of proteins with intrinsically disordered regions. These types of samples pose challenges for other methods, making SAXS an invaluable tool for their characterization.

For folded proteins with available high-resolution structures, BioSAXS utilizes alpha fold structures to discriminate between different models. This unique capability makes BioSAXS less reliant on and complementary to methods like SFX or Cryo-EM.

The comment raised by the reviewer regarding the scope of such measurements and the justification for the difficult experiments and specialized setups remains valid. It is important to clarify that this manuscript aims to evaluate the feasibility of such experiments rather than provide definitive answers. However, it is worth noting that there are dedicated instruments for BioSAXS running or being built in many major synchrotrons. While BioSAXS may be relatively less well-known compared to techniques like macromolecular crystallography or cryo-EM, it has its dedicated practitioners who could be interested in exploring the potential of XFEL SAXS.

The "BioSAXS" part of the introduction has been re-written to better describe the application of the method (lines 71 to 84).

2. The authors cite as an advantage the intense and short pulses of XFELs. I understand the argument about the photons per pulse, but the time-resolution is not used at all in this study. Is the goal (in the long term) to extend the absolute form factor studies to the time-resolved domain? And if so, wouldn't relative changes be enough

to determine photoinduced changes? Would it be really necessary to extract the absolute form factors of photoexcited systems? Why?

Differences in scattering intensities between dark and illuminated state can provide information about the photoinduced change (example in the reference 1 and 2). However, the interpretation is limited and assumptions about the absolute form factor of the molecules are needed. For certain systems, those assumptions can be verified by complementary measurement on Synchrotron SAXS beamline or lab instrument. However, for reaction inducing dissociation/association of molecules, difference curves becomes impossible to interpret and the absolute scattering curves are required.

A paragraph has been added in the discussion for clarification (lines 391-399).

3. Since the low sample consumption needed for the XFEL experiments compared with the synchrotron studies is mentioned as another advantage, would it be possible to give an estimate of the reduction in sample consumption compared to traditional BioSAXS experiments? Isn't this advantage negated by the extremely high jet speeds needed to replenish the jet with the MHz repetition rates used in the experiment?

Providing a precise quantitative estimate of sample consumption reduction between XFEL and synchrotron experiments is challenging, particularly when considering the impact of radiation damage. In synchrotron-based SAXS, sample consumption is fundamentally limited by the effects of radiation damage on the protein, i.e., how much illumination the sample can tolerate before protein aggregation occurs. In contrast, the ultra-short X-ray pulses of XFEL enable overcoming this limitation. With XFEL, all the X-rays within a pulse are scattered by the sample before protein aggregation starts, and subsequent pulses interact with fresh sample. While high jet speeds may raise concerns, a meaningful comparison can be made by considering the number of photons that can be scattered by a 50 μl sample on an XFEL compared to a synchrotron instrument. This provides valuable insights into the potential advantages of XFEL for such measurements. We have incorporated a paragraph in the discussion section (lines 370-384) to address this aspect.

4. On the same note, the authors mention that the sample consumption can be further reduced with the autosampler. I don't follow how that is possible since the jet speed has to be the same. Is the argument here that less data needs to be collected to obtain curves with similar S/N? This is also my issue with Figure 3. The autosampler data has poorer S/N overall (and specially at high Q's) and the authors say that this is due to the "less sample being injected". Ideally one would show data collected with the same number of shots to demonstrate the advantages of the proposed method. I understand this might not be possible anymore due to the nature of beamtime, but one could easily plot the curves obtained by the traditional reservoir method scaled down to the same number of shots as that of the autosampler for a more direct comparison. If this is possible, I would very much like to see this figure in the SI

The argument regarding sample consumption with the autosampler is not based on the signal-to-noise ratio (S/N), but rather on practical considerations for handling small sample volumes. The jet speed and geometry are indeed the same for both the reservoir and autosampler methods.

Regarding Figure 3, a direct comparison of curves obtained with the same number of shots would not demonstrate a better S/N for the autosampler data. In fact, the autosampler data may have a slightly higher S/N due to the dilution of the sample during transport in the jet (as shown in Figure S2). The better S/N observed in the reservoir curves is because all frames of the "sample" reservoir run were averaged, while significantly fewer frames (corresponding to when the sample elutes in the beam) are averaged when using the autosampler.

While the S/N may not be improved in the autosampler data compared to the reservoir data, the advantage of the autosampler lies in its ability to handle small sample volumes more efficiently. The autosampler was specifically developed and optimized for automatically injecting samples into a buffer flow with minimal dead volume and sample cross-contamination. This allows for reduced sample consumption when collecting a SAXS form factor.

To address the concerns raised, the text has been modified (line 291-295) to clarify the origin of the different S/N ratio.

5. Another general comment: A lot of the technical challenges the authors report seem to be specific to high-repetition rate X-ray free electron lasers. I understand that is where they did the experiment but I would recommend that this shouldn't be generalised to all XFELs, since most of the hard X-ray FELs are actually operating in the 30-120 Hz range. Specifically, issues with inter-train intensity variations, jet explosion, fast jet operation and normalization are all non-existent or heavily mitigated when using a low repetition rate source. Have the authors considered performing the same experiments at a different facility? Either the authors should change the title/text to reflect the high-repetition rate XFEL nature of the challenges or comment on the expected performance at other XFEL facilities.

We thank the reviewer for this point and we have updated the discussion (lines 385 - 390) and conclusion (lines 406 - 408) accordingly. The intra-train intensity and pointing fluctuations are lower than the shot to shot variation from the SASE process. The main differences being the jet speed and resulting debris compared to the achievable data rate. We agree that comparable results should be able to be obtained from lower repetition rate sources which can be optimised for the specific conditions at the respective facility. We hope that this work will encourage other experiments to investigate the use of FEL-SAS further at other facilities.

Reviewers' comments:

Reviewer #1 (Remarks to the Author):

In the revised manuscript, the authors enhanced the readability of the manuscript by adding a detailed description of the method they developed, including the schematics of the experimental workflow and data filtering process. In addition, the authors added the discussion on the advantages of the sample delivery system developed in this work in terms of X-ray scattering experiments at XFEL facilities. As a result, the revised manuscript presents the novelty of this study and the experimental details more effectively, compared to the previous one. This reviewer thinks that this revised version provides helpful instructions for future X-ray scattering studies using XFELs and has the potential to attract the interests of the readers. Therefore, the revised manuscript is acceptable for publication in *Communications Biology*. However, before the publication, some suggestions and questions listed below need to be addressed.

1. In line 372, the windowless sample environment is described as an advantage of XFEL-SAS. However, it seems that the windowless sample environment is related to the use of GDVN rather than the use of XFEL facilities. The windowless sample environment can also be achieved at synchrotron facilities if GDVN is used for the experiment in a similar manner.
2. According to the discussion in Line 335 and Figure S4, it was observed that different scattering curves can be made similar through scaling and offset subtraction processes. However, it is essential to perform these processes carefully since the scaling and offset values can influence the molecular shapes determined by the scattering curves. The detailed procedures to determine the proper scaling factors and offset values should be discussed. For now, it seems an arbitrary target scattering curve should be set, and scaling and offset subtraction are performed to match the target scattering curve. If so, this reviewer is doubtful if those procedures would not lead to any artifacts in the resulting molecular shapes.
3. The main topic of this manuscript is the determination of form factors using XFEL-SAS by improving experimental setups and data processing protocols. As such, this reviewer recommends moving figure S6, which summarizes these contents, to the main text.

Reviewer #2 (Remarks to the Author):

The revised manuscript reads well and the added supporting figures are very helpful in understanding and strengthening this manuscript. After reading the revised manuscript and considering the responses from the authors, I recommend publication in *Nature Communications Biology* after addressing some minor comments/suggestions.

1. After the bad scattering images are masked out, it looks like all the integrated scattering curves are overlaid pretty well with proper offsets and scalings (Figure 4s). Can you comment on whether these differences are caused simply by different buffer/air scattering ratios due to sample thickness or something else?
2. Based on lines 376-379, "... an increase of three to four orders of magnitude compared to the typical photon count... in synchrotron SAXS.", I'd wonder if it means that from the photon statistics perspective, XFEL-SAXS has 30-100 times better signal power than synchrotron source given the same experimental conditions. In other words, does it mean that in principle XFEL-SAXS can use 30-100 orders of magnitude lower concentrated samples than the SAXS experiment in synchrotron? Can you comment on this?

3. I'd wonder if the authors could (or tried to) analyze the data obtained by the auto-sampler combined with a chromatogram (Figure 2s) using simple algebra for example. I would expect that one may extract the concentration-independent scattering curve and structure factor.

4. Please make simple fixes for the comments and typos indicated in the attached files.

Reviewer #3 (Remarks to the Author):

I am satisfied with the changes made to the manuscript and the response to my questions. I recommend the manuscript for publication in its current revised format.

Reviewer #1 (Remarks to the Author):

1. In line 372, the windowless sample environment is described as an advantage of XFEL-SAS. However, it seems that the windowless sample environment is related to the use of GDVN rather than the use of XFEL facilities. The windowless sample environment can also be achieved at synchrotron facilities if GDVN is used for the experiment in a similar manner.

The use of GDVN on synchrotrons is indeed being considered, and ongoing tests are being conducted. However, it is important to note that the beam delivered by the XFEL still offers distinct advantages for this type of sample delivery in SAXS experiments. The high coherence of the XFEL beam facilitates the generation of a small beam at both the sample and detector positions, which is essential for accessing X-rays scattered at small angles from micron-sized samples. While it is true that modern synchrotrons, especially 4th generation sources, are increasing in coherence and allowing for improved sample delivery options, the coherence and time structure offered by XFELs remains advantageous in certain cases. Tests are currently underway, including in Hamburg, to evaluate the impact of a windowless environment, considering factors such as sample consumption (longer exposure time) and the engineering challenges associated with achieving similar robustness and reliability as current sample robots used in BioSAXS synchrotron beamline. These evaluations will help determine the feasibility of generalizing this type of setup for broader use.

We mention now the potential use of liquid jet sample delivery in lines 384-390.

2. According to the discussion in Line 335 and Figure S4, it was observed that different scattering curves can be made similar through scaling and offset subtraction processes. However, it is essential to perform these processes carefully since the scaling and offset values can influence the molecular shapes determined by the scattering curves. The detailed procedures to determine the proper scaling factors and offset values should be discussed. For now, it seems an arbitrary target scattering curve should be set, and scaling and offset subtraction are performed to match the target scattering curve. If so, this reviewer is doubtful if those procedures would not lead to any artifacts in the resulting molecular shapes.

We appreciate the reviewer's observation, and we agree that the scaling and offset processes can indeed influence the molecular shapes determined by the scattering curves. The data presented in Figure S4 were collected over a 10-hour beamtime, involving multiple changes in reservoir and jet parameters, leading to differences in the curves. The purpose of showing the scaling and offset was not to correct the curves but rather to demonstrate their overall consistency despite these differences.

Determining precise scaling factors can be challenging. A more reliable method for scaling can be achieved by monitoring the intensity in the beamstop during data collection. Additionally, accurately determining the constant offset requires precise measurements of the jet diameter and its intersection with the beam.

This is precisely to mitigate the potential issues arising from scaling and offset correction, that we propose the use of an autosampler. The autosampler allows for sample and buffer measurements to

be taken within seconds of each other during the same run, thus minimizing the effects of changing beam intensity and jet diameter.

We added clarifications and discuss the possible origin for these discrepancy on lines 323-334 and in the legend of figure S4.

3. The main topic of this manuscript is the determination of form factors using XFEL-SAXS by improving experimental setups and data processing protocols. As such, this reviewer recommends moving figure S6, which summarizes these contents, to the main text.

Thank you for the suggestion, the figure s6 has been moved to the main text

Reviewer #2 (Remarks to the Author):

1. After the bad scattering images are masked out, it looks like all the integrated scattering curves are overlaid pretty well with proper offsets and scalings (Figure 4s). Can you comment on whether these differences are caused simply by different buffer/air scattering ratios due to sample thickness or something else?

The differences observed in the integrated scattering curves, as shown in Figure S4, can be attributed to two main factors: changes in the beam intensity and variations in the liquid jet diameter.

Regarding the scaling factor, the instrument scattering responsible for the upturn at low angles scales with the beam intensity. Therefore, any fluctuations in the beam intensity during data collection would lead to differences in the scaling factor among the curves.

As for the constant offset, it is likely related to the jet diameter. At first approximation, the buffer signal exhibits a flat curve in this scattering range. If we assume that the beam intensity remains constant, curves collected with different jet diameters would have the same overall shape but a small constant offset due to the additional water in the beampath with the larger diameter jet.

In summary, these differences in the integrated scattering curves are likely caused by variations in the beam intensity and liquid jet diameter during data collection. We added clarifications on lines 323-334 and in the legend of the figure S4.

2. Based on lines 376-379, "... an increase of three to four orders of magnitude compared to the typical photon count... in synchrotron SAXS.", I'd wonder if it means that from the photon statistics perspective, XFEL-SAXS has 30-100 times better signal power than synchrotron source given the same experimental conditions. In other words, does it mean that in principle XFEL-SAXS can use 30-100 orders of magnitude lower concentrated samples than the SAXS experiment in synchrotron? Can you comment on this?

Translating photon counts into sample concentration is challenging. All other things equal, an increase of 3 to 4 orders of magnitude in intensity would indeed translate into 30 to 100 times more diluted sample. However, the actual number of photons that can be scattered by, let's say, 50 ul of sample varies based on numerous parameters. For instance, the sample thickness in the capillary

typically used on synchrotron beamlines is better adapted to the typical wavelength used in SAXS (providing an advantage for synchrotron). However, this can lead to sub-optimal use of the sample volume since some part of the sample may not interact with the beam at all. On the other hand, well-adjusted jet sample delivery on XFEL can provide an advantage in this regard. Additionally, radiation damage can limit the quality of BioSAXS curves, defining how many photons the sample can scatter before it starts to degrade and become unusable. XFEL presents a clear advantage here, as radiation damage can be outrun and all photons from the pulses are scattered before the protein aggregates.

To comprehensively assess how XFEL performs in comparison to synchrotron-based BioSAXS for low concentration samples, rigorous experiments on both sources are ultimately required. However, it is essential to recognize that XFEL-based BioSAXS is still in its early stages compared to synchrotron-based BioSAXS, which has been extensively optimized over decades. Direct comparisons may not be entirely fair at this point. Despite this, we firmly believe that XFEL-based BioSAXS offers unique characteristics and advantages that make it an intriguing and promising path to explore for structural studies of low concentration samples

We provided clarifications on lines 398-403.

3. I'd wonder if the authors could (or tried to) analyze the data obtained by the auto-sampler combined with a chromatogram (Figure 2s) using simple algebra for example. I would expect that one may extract the concentration-independent scattering curve and structure factor.

Thank you for the insightful suggestion. Analyzing the data obtained by the autosampler combined with a chromatogram or estimating the concentration during elution based on total scattering intensity could allow us to collect SAXS curves at different concentrations. While our current study was not designed for this specific analysis and focused on obtaining form factors rather than the structure factor, we appreciate the idea and will consider it for a future beamtime.

However, it's important to note that the proposed approach might not fully access the structure factor, especially in the case of repulsive interactions. In such instances, the protein particles could disperse more effectively when injected into the buffer flow, reaching hydrodynamic equilibrium with no significant interaction before being measured by SAXS. This could limit the observation of repulsive interactions and structure factor effects.

To better investigate the structure factor, preliminary experiments using samples with well-characterized structure factors could be conducted on synchrotron beamlines before exploring XFEL experiments.

We are open to discussing this further with the reviewer, but we believe that for the current focus of our manuscript on form factors, it might be best to keep these reflections out of the main text. Thank you for the valuable input, and we will consider this aspect for future research.